# Structure of Photosystem I Supercomplex Isolated from a *Chlamydomonas reinhardtii* Cytochrome b6f Temperature-Sensitive Mutant

**DOI:** 10.3390/biom13030537

**Published:** 2023-03-15

**Authors:** Tom Schwartz, Mariia Fadeeva, Daniel Klaiman, Nathan Nelson

**Affiliations:** Department of Biochemistry and Molecular Biology, The George S. Wise Faculty of Life Sciences, Tel Aviv University, Tel Aviv 69978, Israel; tomschwartz@mail.tau.ac.il (T.S.); fadeevamasha@gmail.com (M.F.); klaiman79@gmail.com (D.K.)

**Keywords:** photosynthesis, *C. reinhardtii*, cytochrome b6f complex, photosystem I, photosystem II, cryo-EM

## Abstract

The unicellular green alga, *Chlamydomonas reinhardtii*, has been widely used as a model system to study photosynthesis. Its possibility to generate and analyze specific mutants has made it an excellent tool for mechanistic and biogenesis studies. Using negative selection of ultraviolet (UV) irradiation–mutated cells, we isolated a mutant (TSP9) with a single amino acid mutation in the Rieske protein of the cytochrome b6f complex. The W143R mutation in the petC gene resulted in total loss of cytochrome b6f complex function at the non-permissive temperature of 37 °C and recovery at the permissive temperature of 25 °C. We then isolated photosystem I (PSI) and photosystem II (PSII) supercomplexes from cells grown at the non-permissive temperature and determined the PSI structure with high-resolution cryogenic electron microscopy. There were several structural alterations compared with the structures obtained from wild-type cells. Our structural data suggest that the mutant responded by excluding the Lhca2, Lhca9, PsaL, and PsaH subunits. This structural alteration prevents state two transition, where LHCII migrates from PSII to bind to the PSI complex. We propose this as a possible response mechanism triggered by the TSP9 phenotype at the non-permissive temperature.

## 1. Introduction

Initially described by Jan Ingenhousz in 1779 as the “great power of purifying the common air in the sun-shine” [1], oxygenic photosynthesis is now known as the multi-step process in which the inorganic materials H_2_O and CO_2_ are converted into organic matter, with O_2_ as a by-product, using the energy of light [2,3,4,5,6]. The ability to use solar energy to power the thermodynamically and chemically demanding reaction of water oxidation, NADP+ reduction, and adenosine triphosphate (ATP) formation is driven by electron transport chains (ETCs), redox interactions, and proton gradients across the thylakoid membrane [5,7,8]. In photosynthesis, several membrane proteins, orchestrated across hydrophobic milieu and hydrophilic environments, work in harmony to catalyze reactions temporally ranging from femtoseconds to days [9,10,11]. 

According to the partial reactions that they catalyze, photosystem II (PSII) is a water-plastoquinone oxidoreductase, the cytochrome b6f complex is a plastoquinone-plastocyanin oxidoreductase, photosystem I (PSI) is a plastocyanin-ferredoxin oxidoreductase, and F-ATPase is a proton-motive force (pmf)–driven ATP synthase [6]. The thylakoid architecture and the configuration of these protein complexes are subjected to constant modulation in response to environmental and physiological stresses [12,13]. However, the sequences of genes encoding the reaction centers (RCs) of the major subunits exhibit very high amino acid sequence conservation [14].

The efficiency of PSI quantum yield has persisted for 3.5 billion years of evolution and has survived an enormous number of potential mutations [5,15,16]. Structural data obtained from species ranging from cyanobacteria to higher plants have demonstrated the strict conservation of the PSI core complex [11,17,18,19,20]. Contemporary organisms contain this functional complex in different but evolutionarily related forms. The subunits PsaA and PsaB are arranged in a central heterodimer, binding the P700 RC and components of the ETC, namely A0 (a monomeric form of chlorophyll [Chl] a), A1 (phylloquinone), and the [4Fe-4S] cluster Fx, forming the PSI core. The Fx cluster is connected to two additional [4Fe-4S] clusters, termed FA and FB, which are bound through the stromal peripheral subunit PsaC [21]. Light-driven charge separation is stabilized within the PSI core via a cascade of rapid electron transfer reactions from P700 via A_0_ and A_1_ to the [4Fe-4S] centers Fx, F_A_, and F_B_. During evolution, subunits were added, and in eukaryotes, a ubiquitous Chl-protein complex (light-harvesting complex [LHC]) emerged [22]. Plant LHCI is composed of four Lhca subunits (Lhca1 to Lhca4), which are arranged in a half-moon-shaped belt that docks on the PsaF pole of the core protein [23]. 

Green algae PSI is very versatile and contains loosely bound LHC subunits. The *Dunaliella salina* PSI was reported to contain only two additional LHCs [24,25,26]. Moreover, a minimal PSI was reported to contain no additional LHCs [26]. The unicellular alga *Chlamydomonas reinhardtii* PSI is much larger than that of higher plants and contains a second half-moon-shaped belt (composed of another four Lhca subunits) and another two Lhca subunits (Lhca2 and Lhca9) on the PsaI pole [27,28,29,30]. The latter have been suggested to play a role in remodeling electron transfer in response to changes in light and/or stress conditions [27,28]. 

Balanced modulation of the excitation energy flux between PSI and PSII is achieved through state transitions [31]. This phenomenon occurs through phosphorylation of LHCII subunits of PSII, resulting in the reversible transfer of the subunits from PSII to PSI (state one to state two transition) [32]. State transitions and their relationship to the kinetic properties of the ETC have been well studied in *C. reinhardtii* [32]. Specifically, the cytochrome b6f complex accumulates in the stroma lamellae in state two [33,34]. A comparative study performed on wild type (WT) and cytochrome b6f mutants tested the theory that in this organism, in state two, most of the excitation energy is utilized by PSI photochemistry. Based on the data, the authors suggested that cyclic electron transport (CEF) around PSI prevails over linear electron flow (LEF) mediated by PSII and PSI [35]. These findings have led to the proposal that in *C. reinhardtii*, modulation of CEF is linked to state transitions. 

In CEF, one electron from PSI needs to be reused by cytochrome bf to achieve an ATP:NADPH ratio of 3:2 in the stroma, in order to assimilate CO_2_ in the Calvin–Benson cycle [36]. Thus, CEF is essential for photosynthetic organisms to run productive electron transport [37]. It has been suggested that when stromal electron carriers are reduced (excess NADPH), the efficiency of CEF is enhanced by the formation of a CEF supercomplex consisting of PSI, cytochrome b6f, and the subunits ferredoxin-NADP-oxidoreductase (FNR), proton gradient regulation-like 1 (PGRL1), anaerobic response 1 (ANR1), and calcium sensor (CAS) [38]. A model has been proposed in which the dissociation of Lhca2 and Lhca9 from PSI supports the formation of this CEF supercomplex [38]. According to the suggested model, the cytochrome b6f dimer associated with PSI-LHCI would bring components involved in stromal electron transfer, the PSI RCs, and ferredoxin binding sites into closer proximity [38]. However, structural evidence for this supercomplex is lacking.

In agreement with the notion that transition to state two requires reduction of the PQ pool, the attainment of anaerobiosis is an absolute requirement to observe the onset of state one to state two transition [32,39]. In most cases, this has so far been reached by using incubation in the dark. In other cases, oxygen evolution has been controlled by medium supplements or material deprivation [40,41]. 

Temperature-sensitive photosynthetic (TSP) mutants, which involve temporary inactivation of processes, have been proposed as tools to elucidate the mechanisms, dynamics, and regulation of light systems [42]. A TSP mutant allows the induction and maintenance of anaerobic conditions in light simply by controlling the culture temperature. A screening method has been developed, and thorough phenotyping of *C. reinhardtii* TSP mutants has enabled the selection of TSP1, TSP2, TSP3, and TSP4, according to a variety of criteria [43]. We applied these phenotyping methods in this study and used TSP4 as an additional control. In TSP4, a single amino acid change (P101H) in the PsbO gene results in a temperature-sensitive PSII. Recently, the structure of PSI from the TSP4 mutant has been solved at 2.54 Å resolution by cryogenic electron microscopy (cryo-EM) [9]. In TSP4, several configurations for the PSI structure have been classified according to the number of LHCs the RC is carrying. PSI-8LHC and PSI-10LHC have been reported [9], in which the omission of Lhc2 and Lhca9 from the complex allows the formation of a PSI dimer [9]. 

In the current study, we selected a cytochrome b6f temperature-sensitive mutant (TSP9), from which we isolated the PSI protein. The TSP9 mutant contained the PSI-8LHC structure and the same LHCs (Lhca2 and Lhca9) were absent in the TSP9 structure. Moreover, in TSP9, there was no trace of the neighboring PsaL and PsaH subunits.

## 2. Materials and Methods

### 2.1. Generation and Isolation of TSP9

#### 2.1.1. Random Mutagenesis

A pool of randomly mutated strains of the model unicellular photosynthetic organism *C. reinhardtii* was produced using ultraviolet (UV) irradiation (Baker company, Sanford, ME, USA), as described previously [43]. UV irradiation of *C. reinhardtii* caused random mutagenesis with a 90% mortality rate (Appendix A and detailed analysis in the Appendix A). Since the single chloroplast of *C. reinhardtii* carries about 75 copies of the genome, newly arising mutations in the chloroplast genome would not be expressed because the WT allele would be in the majority [44].

#### 2.1.2. Negative Selection

We screened roughly 26,000 mutagenized colonies by negative selection for the phenotype of carbon dependency at high temperatures (Appendix A and detailed analysis in the Appendix A). We selected the strains lacking the ability to use photosynthesis for carbon fixation at the elevated temperature of 37 °C. These strains could only be collected from the parallel plates grown at the permissive conditions.

#### 2.1.3. Backcrossing with the WT Strain

To eliminate mutations in their genome that are unrelated to temperature sensitivity, we backcrossed the newly selected mutant strains with the WT, as described elsewhere [43,45]. We screened the progeny again as before, and two daughter cells showed the phenotype out of each tetrad obtained. We selected one, backcrossed it, and screened as above. We repeated this procedure six times for the TSP9 mutant strain.

### 2.2. Photosynthetic Phenotyping: Characterization by Temperature Sensitivity

Temperature sensitivity is characterized by the change in rates of photosynthetic activity measured at permissive and non-permissive temperatures, as shown in Figure 1 and Figure 2 in the Materials and Methods section. We initially grew all cultures at the permissive temperature condition. When needed, we determined the Chl concentration as described elsewhere [46].

#### 2.2.1. Oxygen Evolution Rate (OER) Analysis

We measured the net oxygen exchange rate (O_2_ rate in the light) and the oxygen respiration rate (O_2_ rate in the dark) with a Clark-type oxygen electrode (Oxygraph Plus, Hansatech, Norfolk, UK) [47]. We calculated the oxygen production rates of WT, TSP4, and TSP9 at the three temperature conditions as described elsewhere (see Figure 1) [48]. Graphs were produced using Excel [49]. Statistical analysis of the resulting rates was performed using Graph Pad [50]. Paired *t*-test was performed for “37 °C vs. 25 °C” and “recovery vs. 25 °C” for each strain. Unpaired *t*-test was performed for “TSP4 vs. WT” and “TSP9 vs. WT” for the permitted and non-permitted temperatures. For full statistical analysis, see Appendix A.

#### 2.2.2. PSII Activity by Fluorescence Measurements

We recorded fluorescence absorbance by using a Joliot-type spectrophotometer (JTS-10; BioLogic, Grenoble, France). We calculated the maximum quantum efficiency of PSII (Max qe) for each strain at the three temperature conditions (colors coded as before) as the F ratio, (Fm − F0)/Fm, as described previously (see Figure 2) [48,51]. Graphs were produced using Excel [49]. Statistical analysis of the resulting rates was preformed using Graph Pad [50]. Paired *t*-test was performed for “37 °C vs. 25 °C” and “recovery vs. 25 °C” for each strain. Unpaired *t*-test was performed for “TSP4 vs. WT” and “TSP9 vs. WT” for the permitted and non-permitted temperatures. For full statistical analysis, see Appendix A.

#### 2.2.3. Western Blot Analysis

We cultured WT *C. reinhardtii* and the sixth generation of the TSP9 mutant in TAP+ medium (Tris, acetate, phosphate at pH 7.0) [52,53] at 25 °C, adapted to 37 °C for 14 h, and subsequently recovered at 25 °C for 24 h. We isolated thylakoids from each of the three conditions and then lysed the samples with sodium dodecyl sulfate (SDS) dissociation buffer. We used SDS-polyacrylamide gel electrophoresis (PAGE) with a 17% gel to separate the proteins and then transferred the proteins to a cellulose membrane using a semi-dry transfer method (Trans-Blot SD, Bio-Rad, Santa Clara, CA, USA), according to the manufacturer’s instructions. Finally, we subjected the membranes to immunoblotting [54]. The amount loaded corresponded to a Chl content of 1.0 µg (see Figure 3 in the results section and Appendix A and detailed analysis in the Appendix A).
Each of the different complexes was represented by a few subunits (three for PSI, five for PSII, two for cytochrome b6f, and two for ATPase). Therefore, we compared proteins present in the isolated membranes taken from cells grown at the three temperature conditions (see Appendix A).

### 2.3. Genotyping TSP9 Mutant

#### 2.3.1. Sequencing and Gene Analysis

We amplified the nuclear genes coding for the cytochrome b6f complex proteins (petO, petN, petM, and petC) of the WT strain and the two daughter cells of the sixth generation of TSP9 (a phenotypic progeny and a non-phenotypic progeny) by colony polymerase chain reaction (PCR, ESCO Healthcare Swift MiniPro, Changi South, Singapore) and then analyzed the products with Sanger sequencing. We aligned the sequences to the *C. reinhardtii* DNA database and compared them to locate mutations in the genes (see Appendix A).

#### 2.3.2. Mutant Protein Stability Analysis

We examined the effect of the detected mutation by using the DUET and Dynamute online servers for protein stability predictions [55,56]. The first server uses machine learning algorithms, and the second uses normal mode analysis [57,58]. The two predictions express the protein stability as the variation in Gibbs free energy (∆∆G) caused by the amino acid substitution. The results are described in Section 3.3. The full results of the stability predication analysis are given in Appendix A.

#### 2.3.3. Multiple Sequence Alignment (MSA) for Amino Acid Conservation Survey

We aligned the amino acid sequence by using the BLAST protein database [59] and indeed retrieved similarities to the petC chain in various photosynthetic organisms, from algae to higher plants. We chose the sequences of green and red algae for alignment and observed the conservation levels of the amino acids near the mutation site in all closely related sequences found for this gene. We used 13 sequences as input for MSA analysis by the PRALINE server [60]. The full results of the MSA analysis are given in Appendix A.

### 2.4. Electron Transport Analysis

#### Kinetics of Light-Induced P700 Oxidation and Reduction

We grew the WT and TSP9 strains in 1 L of liquid TAP+ medium at 25 °C with shaking, and under continuous illumination (30–45 μE/m^2^s). We measured all light intensities using an LI-250A light meter (LI-COR, Lincoln, NE, USA). For each of the strains, we divided the cultures equally into three conditions and added TAP+ to reach a final volume of 1 L. For the permissive temperature condition, we grew the strains at 25 °C for 24 h, while for the non-permissive condition, we grew the strains at 37 °C for 24 h. Finally, for the recovery condition, we transferred heat-treated samples back for an additional 24 h at the permissive temperature of 25 °C. With the initial inoculations and dilutions, we aimed to reach a logarithmic growth stage (measured as an optical density [OD] at 730 nm of 0.2–0.7) at the time of measurement. On the day of the measurements, we harvested 500 mL of the cultures by centrifugation (Sorvall F10S 4x1000 LEX rotor, Thermo Fisher Scientific, Waltham, MA, USA; 6800× *g* for 8 min). We resuspended the pellet into 5 mL of the medium extracted from the same culture. We measured the Chl content of the concentrated cultures and diluted all samples (using their respective extracted mediums) to reach the same Chl content of 0.1 mg/mL. Then we mixed 1 mL of each sample and measured light-induced oxidation of P700 to P700+ and re-reduction of P700+ to P700. This was done by recording absorption changes at 705 nm using a JTS-10 spectrophotometer [25]. We measured the strains’ reactions to varying temperatures in the absence and presence of 2,5-dibromo-6-isopropyl-3-methyl-1,4-benzoquinone (DBMIB), an inhibitor of PQ-oxidation by cytochrome b6f [61,62].

### 2.5. Structural Data Collection and Analysis

#### 2.5.1. PSI Isolation

We grew *C. reinhardtii* TSP9 in 10 L of TAP+ medium for 5 days at 25 °C under continuous illumination (35–45 µE/m^2^s). We then transferred the culture to 37 °C and left it to adapt for 20 h under continuous illumination (35–45 µE/m^2^s), reaching an OD at 730 nm of 0.7. We harvested the culture by centrifugation (Sorvall F10S 4x1000 LEX rotor; 10,000× *g* for 5 min) for thylakoid membrane preparation. We resuspended the pellet and washed it in 100 mL of STN buffer (0.3 M sucrose, 30 mM Tricine-Tris pH 8.0, and 10 mM NaCl) and centrifuged it again (Sorvall SS-34 rotor; 11,952× *g* for 10 min). We then resuspended the pellet in 30 mL of ice-cold STN buffer and added protein inhibitors (1 mM PMSF, 2 mM benzamidine, 1 µM Pepstatin Sigma cocktail, and 50 µM BST). We broke the cells by using a French press (Avestin^®^ EmulsiFlex-C3 electric motor; three times at 1500 psi). We removed unbroken cells and starch by centrifugation (Sorvall SS-34 rotor; 12,000× *g* for 10 min) and discarded them. We centrifuged the supernatant at a higher speed (Beckman Ti-70 roto; 210,000× *g* for 1.5 h) to recover the thylakoid membranes. We resuspended the pelleted membranes and washed them in 80 mL of STN buffer, diluted to a Chl concentration of 1 mg/mL, solubilized with 1.5% of *n*-decyl-β-D-maltoside (β-DM), and incubated them in the dark on ice for 15 min. We removed insoluble material by centrifugation (Beckman Ti-70 rotor, Beckman Coulter International S.A., Nyon, Switzerland; 210,000× *g* for 20 min). Next, we centrifuged the supernatant (Beckman Ti-75 rotor; 225,000× *g* for 20 h) and resuspended the pellet in 10 mL of STN 0.1 buffer supplemented with 0.5% β-DM (0.1 M sucrose, 30 mM Tricine-Tris pH 8.0, 10 mM NaCl, and 0.5% β-DM) and centrifuged it in a bench-top centrifuge (20,800× *g* for 5 min). The supernatant Chl content was 4 mg/mL, and we loaded 0.5 mL of it (containing 2 mg of Chl) onto a sucrose gradient (20 mM Tricine-Tris pH 8.0, 0.2% β-DM, and 10%–45% sucrose). We centrifuged the gradients (Beckman SW-40 rotor; 240,000× *g* for 16 h) and collected the thick band appearing the next day as three layers (low, middle, and upper). For each layer, we determined the Chl concentration and measured the P700 kinetics. The lower band detected on the gradient contained the PSI protein and had a Chl concentration of 0.7 mg/mL. We loaded the sample on a 50 mL DEAE anion-exchange column that was previously equilibrated with TN buffer (30 mM Tricine-Tris pH 8.0 and 10 mM NaCl) containing 0.2% α-DM. We washed the column with the same buffer containing 50 mM NaCl and eluted the particles with the same buffer containing 200 mM NaCl. We loaded the elution fractions with the highest Chl concentration on a second sucrose gradient and centrifuged it (Beckman SW-60 rotor; 435,000× *g* for 4 h). We collected the lowest band, precipitated it with PEG [63], and resuspended the pellet in a buffer containing 20 mM Tricine (pH 8.0) to a Chl concentration of 11.152 mg/mL. Then, we diluted the sample in the same buffer to reach a Chl concentration of 4 mg/mL. We used a JTS-10 spectrophotometer to measure samples taken from purification steps for P700 kinetics to demonstrate the PSI concentration in the sample. We dissolved the same samples and separated the proteins with SDS-PAGE, followed by Coomassie staining the gel to show the protein purification process. The corresponding P700 is given in Appendix A for each purification step.

#### 2.5.2. PSII Isolation

We grew *C. reinhardtii* TSP9 cells in 10 L of TAP+ medium. We cultured the cells with constant stirring and air bubbling under continuous white light (35–40 μE/m^2^s) at 25 °C for 24 h. Afterwards, we transferred the culture to 37 °C (with the same air bubbling and stirring) for 16 h. After the heat treatment, we checked the OER of the culture (final OD at 730 nm of 0.84) and harvested the culture by centrifugation (Sorvall F10S 4x1000 LEX rotor; 10,000× *g* for 5 min) and suspended it in wash medium (25 mM HEPES-NaOH pH 7.0, 300 mM sucrose, and 5 mM MgCl_2_) [64]. We washed the cells once in the same buffer, spun them down by centrifugation (Sorvall SS-34 rotor; 6500× *g* for 5 min), and suspended the pellet in breaking buffer (25 mM Mes-NaOH, pH 6.0, 1 mM MgCl_2_, 10 mM NaCl, 1 M betaine, and 200 mM sucrose) [64]. We added a protease inhibitor cocktail (1 mM PMSF, 2 mM benzamidine, 1 µM Pepstatin Sigma Cocktail, and 50 µM BST). We disrupted the cells with an Avestin EmulsiFlex-C3 at 2000 psi (three cycles). We removed unbroken cells and starch granules by centrifugation (Sorvall SS-34 rotor; 12,000× *g* for 10 min) and precipitated the membranes in the supernatant by centrifugation (Beckman Ti-70 rotor; 160,000× *g* for 45 min). We resuspended the pellet in the same breaking buffer, giving a Chl concentration of 2 mg/mL. We added α-DM and *n*-octyl β-D-glucopyranoside dropwise to a final concentration of 2% and 1%, respectively, to a final Chl concentration of 1.5 mg/mL. After stirring at 4 °C for 30 min, we removed the insoluble material by centrifugation (Sorvall SS-34 rotor; 26,900× *g* for 10 min). We loaded the supernatant (≈800 µg of Chl per tube) on a sucrose gradient (0.1% αDM, 25 mM Mes-NaOH, pH 6.0, 1 mM MgCl_2_, 10 mM NaCl, 1 M betaine, and 20–50% sucrose). We centrifuged the gradients (Beckman SW-60 rotor; 310,000× *g* for 15 h) and measured the OER of each layer appearing on the gradient. These data are presented in Appendix A, with the corresponding protein levels demonstrated by Coomassie blue staining of the SDS-PAGE gel. We diluted the layer containing PSII (marked layer C in Appendix A) to reduce the sucrose concentration with breaking buffer containing αDM 0.1% and concentrated it by using Vivaspin^®^20 (MWCO 100,000 PES membrane). We loaded the concentrated preparation on the same sucrose gradient and ran it at the same conditions as before.
We collected the middle of the resulting green band, diluted it to remove sucrose with the breaking buffer containing 0.1% αDM, and concentrated it using Vivaspin^®^20 (MWCO 100,000 PES membrane) to give a Chl concentration of 0.8 mg/mL. The final preparation exhibited an OER of 152 µmol O_2_/mg Chl/h under 510 µmol photons/m^2^s illumination (the steps and resulting OERs are given in Appendix A).

#### 2.5.3. Cryo-EM Data Collection and Processing

We applied the purified PSI sample (3 μL) onto glow-discharged holey carbon grids (Cu Quantifoil R1.2/1.3) before vitrifying using a Leica-EM-GP (3 s blot at 20 °C and 90% humidity). We collected images using a 300 kV FEI Titan Krios electron microscope, with a slit width of 20 eV on a GIF-Quantum energy filter, at the ESRF Cryo Facility, Grenoble, France. We used a Gatan K3-Summit detector in counting mode at a magnification of 130,000 (yielding a pixel size of 0.84 Å), with a total dose of 50.4 e/Å^2^. We used EPU to collect a total of 13,173 micrographs, which we dose fractionated into 50 video frames, with defocus values of 0.5–9.0 μm at increments of 0.5 μm. We motion-corrected and dose-weighted the collected micrographs using MotionCor2 [65]. We estimated the contrast transfer function parameters by using CtfFind v.4.1 [66]. We picked a total of 1,009,378 particles using the LoG reference-free picking in RELION3 [67]. We processed the picked particles for reference-free two-dimensional (2D) averaging. After several rounds of 2D classification, which resulted in 403,290 particles, we generated an initial three-dimensional (3D) model by using RELION3. We pooled a total of 173,187 (TSP9-PSI-8LHC) particles together and processed for 3D homogeneous refinement and post-processing with RELION. The reported resolutions are based on a gold-standard refinement, applying the 0.143 criterion on the FSC between the reconstructed half-maps (the flow chart of model building and related images are shown in Appendix A).

#### 2.5.4. Model Building and Processing

To generate the TSP9-PSI, we selected the cryo-EM structure of *C. reinhardtii* PSI model PDB 6JO5. We fitted this model onto the cryo-EM density map and manually rebuilt it using Coot [68]. We performed stereochemical refinement using phenix.real_space_refine in the PHENIX suite [69]. We validated the final model using MolProbity [70]. The refinement statistics are also provided in Appendix A. We determined local resolution with RELION local resolution [67] and generated the figures using PyMOL [71] and UCSF Chimera [72].

## 3. Results

### 3.1. Generation and Isolation of TSP9

In this study, we used a selection system designed to identify mutations in the photosystems (see Section 2), as described elsewhere [43]. When exposed to light, the WT strain of *C. reinhardtii* produces biomass and chemical energy. We attempted to generate a mutant strain that was sensitive to temperature changes. To achieve this, we genetically manipulated the WT strain by inducing random mutagenesis in its genome. After producing a pool of potential new strains, we screened them via negative selection for temperature sensitivity. Once we identified mutant strains with the desired phenotype, we measured and evaluated their oxygen evolution rates. We tested mutant strains with reduced oxygen rates at the non-permissive temperature for their performances and ability to recover oxygenic activity. TSP9 is one of 17 TSP mutants we selected and characterized by photosynthetic phenotyping for temperature sensitivity. To eliminate mutations unrelated to the photosynthetic phenotype, we backcrossed TSP9 with the WT strain six times. Related single nucleotide polymorphism (SNPs) [48].

### 3.2. Photosynthetic Phenotyping: Characterization by Temperature Sensitivity

Initially, we characterized TSP9 by measuring OERs and PSII activity by fluorescence (Figure 1 and Figure 2, respectively). We used WT and TSP4 cultures as controls. For all strains, the blue bars correspond to the permissive temperature condition (25 °C), the red bars to the non-permissive temperature condition (37 °C), and the gray bars to the recovery condition (24 h at 25 °C, after 24 h at 37 °C).

The oxygen production rate of the WT strain did not change significantly in response to the change in temperature (the O_2_ production rate was 159 ± 42 nmol O_2_/mg Chl per min at 25 °C, 139 ± 22 at 37 °C, and 134 ± 34 for recovery). For the TSP mutants, the reduction in oxygen production rates in response to temperature change was significantly stronger (Figure 1 and Appendix A)). TSP9 repeatedly showed a reduction in oxygenic activity and recovery of all photosynthetic apparatuses and functions once returned to the permissive temperature condition.

Unlike the TSP4 mutant, the TSP9 mutant showed a small difference from the WT in PSII activity in response to a high temperature. The significance was found at a *p*-value of 0.0032 (Figure 2 and Appendix A).

We used representative subunit-specific antibodies to examine the effects of elevated temperature on the presence of the PSI, PSII, cytochrome b6f, and ATP synthase complexes in isolated thylakoids from the WT and TSP9 strains (full western analysis can be found in Appendix A). The immunoblot signals indicated no significant change in protein levels of the WT strain in response to temperature. The TSP9 mutant showed signals for all four photosynthetic complexes at the permissive temperature (25 °C). While there was some reduction in the levels of the petC (Rieske protein) signal, the levels of the cytF (petA) polyclonal IgG were similar to those of the WT strain. However, when adapted to the non-permissive temperature (37 °C), the TSP9 mutant showed a significant decrease in the level of cytF and complete elimination of the petC signal (Figure 3). PetC and cytF exhibited recovery upon return to the permissive temperature.

The OERs at 25 and 37 °C (Figure 1), fluorescence signals for PSII activity (Figure 2), and the loss of specific complexes signals in the immunoblot analysis (Figure 3) of TSP9 suggest that the mutation most likely occurred in one of the nuclear genes encoding a cytochrome b6f complex protein.

We analyzed TSP9 growth rates in TAP+ (mixotrophic) and TAP- (photoautotrophic) medium based on the Chl concentration [46], OD at 730 nm, and cell viability counts, measured over time (Appendix A). We performed these growth tests using both the WT and TSP4 strains to compare TSP9 survival and resilience with these known strains. TSP9 growth was better than TSP4 and as good as WT growth, even at the non-permissive temperature when supplied with carbon. For all strains, photoautotrophic growth (in the TAP- medium) was much slower than the rates observed when carbon was supplied to the culture. The differences in the growth rates of the WT and mutant strains at the permissive temperature were relatively minor. However, at the non-permissive temperature, the TSP strains showed a negative growth curve over time, representing their inability to assimilate carbon via photosynthesis. We expected this outcome because we specifically selected these mutants for their temperature-dependent inactivation of photosynthetic growth.

### 3.3. The TSP9 Phenotype Results from the W143R Mutation in petC of the Cytochrome b6f Complex

We sequenced and analyzed the nuclear genes encoding cytochrome b6f complex proteins of the WT strain and the sixth generation of TSP9. There were no changes in the petO, petN, and petM genes. Alignment of the resulting sequences identified the mutation in the petC gene, which encodes the Rieske protein (Appendix A). There was a single substitution of T to C in exon 3 of petC, as seen in the alignment of the mutant sequence against the WT *C. reinhardtii* gene [73]. This SNP did not appear in the sequences of the WT strain or the non-phenotypic progeny of TSP9. This single nucleotide change caused a single amino acid replacement at position 143, substituting the aromatic non-polar amino acid tryptophan (W) with a positively charged amino acid arginine (R). The translated sequence of amino acids for this exon at the mutation site is: 133-VCTHLGCVVP**W**VAAENKFKCP-153. The mutated tryptophan (W) is underlined. We generated a model of this single amino acid substitution by using COOT local refinement [68] and examined the potential electrostatic interactions with the neighboring amino acids (Figure 4).

The 2Fe-2S cluster of the protein is ligated by the H136, C134, H155, and C152 residues, situated downstream and upstream of the mutation site. The mutation is located on a β-sheet strand near the 2Fe-2S cluster, just four amino acids downstream of a cysteine (C139) residue that forms a crucial disulfide bond (with C154).

The protein stability predictions gave a value of −1.59 kcal/mol based on the DUET server [55] and −1.64 kcal/mol based on the Dynamute2 server [56]. Negative values denote destabilizing mutations [57,58]. (For the full resulted predictions, see Appendix A.)

The MSA showed that this tryptophan is highly conserved in all sequenced species of green and red algae. The tryptophan was graded at a conservation level of 9/10. In some higher plants such as spinach, it was replaced by phenylalanine (F), which is also an aromatic non-polar amino acid. In none of the cases, it was replaced by a charged amino acid (Appendix A and detailed analysis in the Appendix A).

### 3.4. Effect of TSP9 Mutation on Electron Transport

Similar to the approach used to study the plastocyanin (PC) concentration [74,75], we followed the P700 oxidation and reduction levels in intact cells as an indicator of the functionality of the cytochrome b6f complex (Figure 5).

The time course of the absorption changes in P700 recorded by the JTS-10 spectrophotometer in the dark-adapted algae represents the rate of reduced P700+/oxidized P700 in the sample after a flash of light. When the culture’s Chl concentrations are equally balanced, the absorbance signals could indicate the state of electron flow around PSI. In vivo, upon light excitation of P700, the excited electrons in PSI transfer according to their mid-point redox potentials (Em) to A_0_, PhQ, Fx, F_A_, and F_B_, which eventually reduce Fd. PC serves as the electron donor to P700+. The rate constant for this electron transport depends on the components’ affinities to electrons [76]. Thus, the rate of oxidized P700 equilibrium depends on the components midpoint redox potential, their concentration in the sample, and the membrane potential [77]. The bonding and the disassociation of PC to P700 affect the midpoint redox potential of the complexes [78]. Under standard conditions, LEF funnel electrons from PC to P700. The initial oxidation occurs in less than 1 µs; most of it is linked to PSI activity, and the other 20–40% is related to PSII activity [79]. Lower absorbance signals, such as −1400 relative units for TSP9 cells at 37 °C, indicate more oxidized P700 recorded relative to those of TSP9 grown at the permissive temperature (−600 relative units). In the following phase, slow oxidation is derived from PSI, and when the light is turned off, slow reduction is observed. The difference in absorbance measured indicates a change in the ratio of P700+ reducers (electron donors) to P700 oxidation levels (by electron acceptors) during illumination. Because we performed the measurements in vivo, as oxidized PC is in excess after the light turns off, the small equilibrium constant, and the long half-time of its disassociation from P700, limits electron transfer [78]. We interpretated implications concerning the LEF status of the cells by using the recorded kinetics. When the light was turned off, there was additional P700 oxidation in all recorded measurements, including the WT. We assumed that this oxidation represented the back electron transfer from P700 to oxidized PC [78], and therefore we chose not to focus on it.

The WT cells exhibited similar kinetic properties regardless of the temperature change. The TSP9 cells grown at 25 °C showed similar kinetic properties when compared with the WT cells. However, when adapted at 37 °C for 24 h to eliminate the cytochrome b6f complex, the TSP9 cells showed a significant change in P700 photooxidation level, likely due to the lack of electron flow from PSII through PC when illuminated.

DBMIB prevented electron transfer from the cytochrome b6f complex to PC, decreasing P700+ reduction of the WT cells, which can be seen as a stronger oxidation signal in the light and slower reduction in the dark. The light-induced P700 signal of TSP9 cells adapted to 37 °C was very similar to those of the WT cells in the presence of DBMIB.

The P700 kinetics of TSP9 shown in Figure 5 indicate the lack of electron flow from PC. When the Rieske protein cannot donate electrons, both PC and P700 remain oxidized. When the Rieske protein cannot accept electrons, PQH_2_ remains reduced. There were no changes in the core PSI subunits under the above conditions (based on the immunoblot analysis, see Appendix A). Thus, we set out to solve the structure of PSI in the TSP9 strain by using cryo-EM to determine whether there was a change in the peripheral subunits.

### 3.5. Structures of Photosynthetic Complexes Isolated from the TSP9 Mutant Grown under Nonpermissive Temperature

#### 3.5.1. Cryo-EM Structure of the PSI Complex

The PSI structure isolated from TSP9 at the non-permissive temperature is shown in Figure 6. It is apparent that the PsaL and PsaH subunits, as well as Lhca2 and Lhca9 are absent.

The structure of TSP9 PSI-8LHC suggests that stress resulting from the lack of the cytochrome b6f complex modulated the structure by controlling the number of attached LHC subunits and the lack of subunits necessary for state II transition.

The high resolution obtained for this structure reflects the homogeny of the sample particles. Indeed, the majority of the particles showed this conformation; however, we noted a small number of particles with a large PSI complex during particle classification. These large PSI particles occurred at a very low frequency in the sample and provided insufficient structural data to solve a structure. The high-resolution structure of TSP9 PSI is in agreement with previous reports of this configuration and adds to the reliability of the suggested mechanism for membrane response to stress. In Figure 7, the local resolution of the obtained structure is mapped by color.

#### 3.5.2. Cryo-EM Structure of the PSII Complex

We isolated PSII protein and then determined its structure at a low resolution. Although we collected data from a sufficient number of particles, we were not able to solve it to near-atomic resolution. So far, we have not identified significant structural alterations in the TSP9 PSII supercomplex.

## 4. Discussion

The use of photosynthesis-inhibiting or -inducing compounds to study photosynthetic dynamics is common, and many of the materials involved are well-known and understood. In addition, a vast array of photosynthetic mutants is available. These substantial efforts have been made to break down the multistep process of photosynthesis into its different elements so that the relative contribution of each protein to the ETC can be better understood. However, temporary inhibition by materials and conditional recovery mutants are not as commonly used. TSP9 is one such mutant.

In our previously published paper characterizing the TSP4 mutant, we reported a structured network of water molecules present in the PSI core complex. Additionally, we found that the structures of the LHCs had been conserved in cyanobacteria, green algae, and higher plants [9]. These findings demonstrate the strict conservation of the PSI core complex [10,11,17,19,20] and its modularity and flexibility in adapting to countless environments and conditions [25]. Over the years, many regulatory and adaptative mechanisms involved in photosynthesis have been uncovered, some of which we have discussed here. The observation of conserved ligand positions (water in the case of PSI) serving as protein conformation modulators (of the electrostatic landscape of the ETC in the case of PSI) and PSI oligomerization, first reported in eukaryotes, both demonstrated by a temperature-sensitive mutant, highlights the ability to produce high-resolution structures from photosynthetic mutants as a unique tool for analyzing proteins under conditional stress.

The negative selection approach, where the desired mutant does not grow under the restricting condition, is rarely used and has seldom been exploited to study photosynthesis [43,80]. Yet, this tedious “negative selection” has yielded mutants that would have been otherwise overlooked [9]. Here, we used negative selection to isolate a TSP mutant in the cytochrome b6f complex of *C. reinhardtii*, which we refer to as TSP9. Unlike most of the TSP strains previously tested, TSP9 regained oxygen production efficiently when returned to the permissive temperature and showed a longer survival time than other TSP mutants, including the four published ones [43].

It is reasonable to assume that the mutation caused a conformational change in the landscape of the Rieske protein, which resulted in structural destabilization. In the WT strain, residue R104 is closely packed against residue W143, and there are electrostatic interactions between the positive charge of the arginine residue and the negative partial charge of the π electrons of the tryptophan residue. The mutated protein introduces another positively charged amino acid instead of the tryptophan and thus nullifies the stabilizing interactions and adds electrostatic repulsion between the two arginine residues. We expected that these two electrostatic penalties destabilize the protein’s structure.

Somewhat similar phenotypes have been observed in *Chlamydomonas* petC and petD site-directed mutants [34,81]. Levine and Smillie [82] produced mutations of deletions and substitutions in petC. In the ac21 mutant, Vitry et al. [34] reported a W163R substitution (according to current datasets, it should be referred to as W192R) that led to reduced accumulation of the Rieske protein. The authors suggested that this mutation causes a modification in the functional organization of the thylakoid membrane. The specific mutation of TSP9 is located 49 amino acids upstream of the ac21 mutation site. Importantly, ac21 and the other mutant strains reported did not demonstrate oxygenic activity, leading to the conclusion that certain aromatic residues are necessary for Rieske protein assembly and stability.

The cytochrome b6f complex is involved in state transitions, as shown by their absence in *C. reinhardtii* mutants lacking this complex [83,84]. Moreover, cytochrome b6f has been suggested to enhance CEF by direct association with PSI [38]. It has also been proposed that the reversible dissociation of the Lhca9/Lhca2 heterodimer from the PSI core may provide a docking site for the cytochrome b6f complex [28,38]. The temporary loss of the Rieske protein in the TSP9 strain, which causes the cytochrome b6f complex to lose its activity, appears to be an interesting tool to investigate these models. We were able to grow large volumes of TSP9 cultures in a mixotrophic manner at the permissive temperature and then transfer the cultures to the non-permissive temperature to inhibit LEF via the lack of cytochrome b6f activity.

As the cytochrome b6f complex plays an important role in both CEF and LEF [34,38,62,85,86], the ability of the TSP9 strain to link its activity or lack of activity with other factors involved in the electron chains is a unique advantage. By mating the TSP9 strain with other *C. reinhardtii* mutant strains, the temperature-dependent activity can be transformed, allowing the conditioning of other strains to reveal parts of the reaction that might have escaped detection so far. 

The observed differences in P700 photooxidation for TSP9 cells adapted to 37 °C suggests a change in either the composition of PSI or its interaction with other potential electron donors as P700 was eventually reduced. Although at a slower rate, electrons were still delivered to PSI, even when the Rieske protein had degraded and the primary electron acceptor (PC) was unable to receive electrons from the cytochrome b6f complex. These P700 kinetics and the lack of O_2_ production demonstrate that LEF of the TSP9 strain at the non-permissive temperature is inhibited. This would keep plastoquinone (PQ) reduced, which has been shown to induce state transition [32]. Growing the TSP9 strain at the non-permissive temperature should enable state transition. It could be expected that the structure produced would show PSI under state two. The PSI protein of TSP9 is not mutated, so the structural alteration can be attributed to the response to stress. Hence, we were able to isolate the PSI structure from the TSP9 strain. The structure showed that the mobile LHC of the TSP9 strain could not attach to PSI because it lacks the PsaL and PsaH subunits, which are responsible for binding LHCII in state two. Most likely, in the absence of the cytochrome b6f complex, mutant cells are fixed in one of two alternative states. Without PsaH and PsaL, the only possible state for TSP9 is state one. The structural response seen here might prevent the over-excitation of PSI and thus protect the protein from phototoxicity. Another possible reason for this conformational response may stem from the production of reactive oxygen species (ROS) by PSII when LEF is inhibited.

This structure concurs with previous findings of mutant plants lacking PsaH and PsaL subunits with deficient state transition [87]. These subunits are placed on the opposite side of the LHCI belts and have previously been hypothesized to constitute a binding site for LHCII [88]. Very recently, it has been established that PsaH is required for the lateral binding of LHCII to the PSI core in state transitions. It has been suggested that PsaH-lacking complexes represent a previously overlooked functional form of PSI, where PsaH is either downregulated or unable to be assembled, and this may further reflect regulation of PSI dimerization [89].

The TSP9 mutant provided an opportunity for us to examine the structural dynamics of photosynthetic proteins, and the observation of another case of a small PSI structure allows for a deeper understanding of the plasticity and possible response of the photosynthetic apparatus to environmental changes. The structural alterations could be a response to the anaerobic conditions triggered by the TSP9 strain, but could also be a response to the temperature change. Because no structure had previously been solved for the WT strain at the non-permissive temperature, the latter should not be ruled out. However, in the TSP9 strain, the loss of Lhca2 and Lhca9 means that the PSI–cytochrome b6f supercomplex cannot assemble under these conditions. Given the unstable binding of Lhca2 and Lhca9 to PSI, it is possible that these subunits were lost during the purification process. However, because the isolation process did not differ from protocols previously used for WT cells, it is more likely that the subunits had weaker binding due to the temperature or the anaerobic conditions.

## 5. Conclusions

Under a restrictive temperature, the cytochrome b6f complex is practically lost in the TSP9 strain. Under these conditions, cells must cope with a lack of electron supply to photooxidize P700. Our structural data showed that the mutant responded by lack of assembly of PsaL, PsaH, and two LHC subunits (Lhca2 and Lhca9) in the PSI supercomplex. This structural alteration not only reduces the excitation landscape of PSI but also prevents state II transition, where LHCII migrates from PSII to bind the PSI complex. The structure of small PSI seen in the TSP9 strain supports recently published descriptions regarding PSI complex conformations [9,89] as a possible response to stress.

## Figures and Tables

**Figure 1 biomolecules-13-00537-f001:**
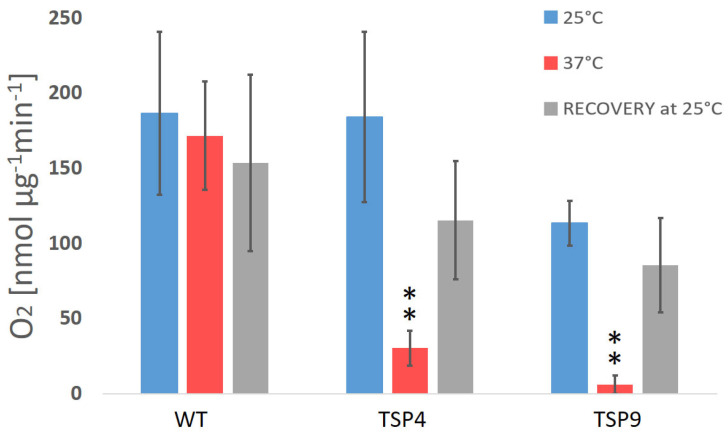
Average rate of O_2_ production (in nmol O_2_ per µg chlorophyll for 1 min) in *Chlamydomonas reinhardtii* wild type and TSP mutant at three temperature conditions (color coded). The oxygen production rates of the wild type (WT), TSP4, and TSP9 mutants were measured at a light intensity of 300 μE/m^2^s and in the dark and calculated relative to the corresponding chlorophyll levels. Graphs were produced using Excel. Statistical analysis was performed using Graph Pad. The data represent the mean ± standard deviation of five to eight independent experiments, outliers were removed. The significance level by an unpaired *t*-test for “TSP4 vs. WT” and “TSP9 vs. WT” at 37 °C is indicated as follows: no sign means *p* > 0.01 and ** means *p* < 0.001.

**Figure 2 biomolecules-13-00537-f002:**
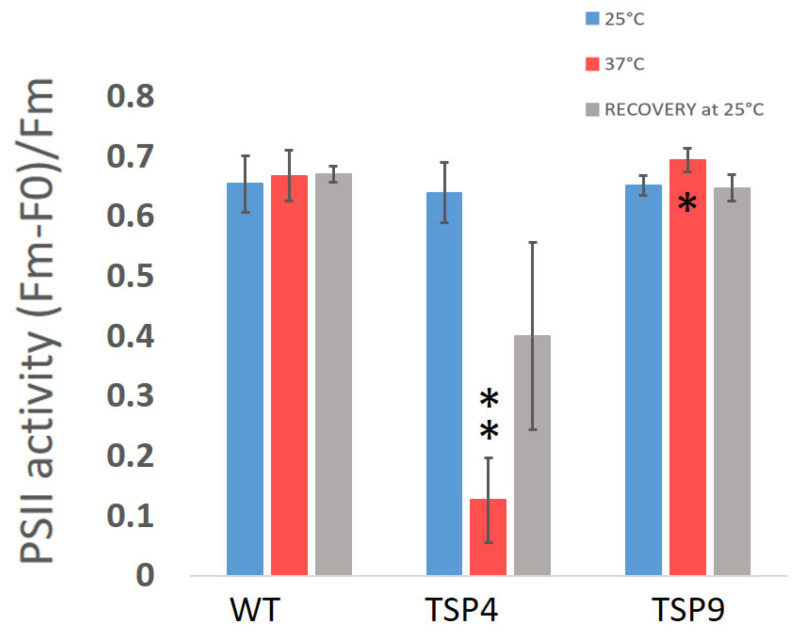
Average ratio of the maximum quantum efficiency of photosystem II (PSII) in the *Chlamydomonas reinhardtii* wild type (WT), TSP4, and TSP9 strains at three temperature conditions. Graphs were produced using Excel. Statistical analysis was performed using Graph Pad. The data represent the mean ± standard deviation of six to seven independent experiments. The significance level by an unpaired *t*-test for “TSP4 vs. WT” and “TSP9 vs. WT” at 37 °C is indicated as follows: no sign means *p* > 0.01, * means *p* < 0.01 and ** means *p* < 0.001.

**Figure 3 biomolecules-13-00537-f003:**
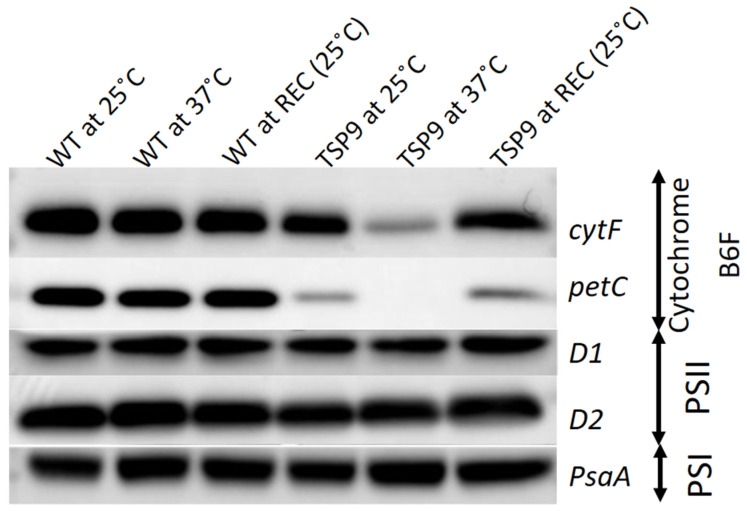
Immunoblots of isolated thylakoid membranes from the wild type (WT) and TSP9 strains grown at three temperature conditions (indicated at the top). The labels to the right of the panels indicate the various subunits (left column) and the complex to which they belong: cytochrome b6f, photosystem I (PSI), and photosystem II (PSII).

**Figure 4 biomolecules-13-00537-f004:**
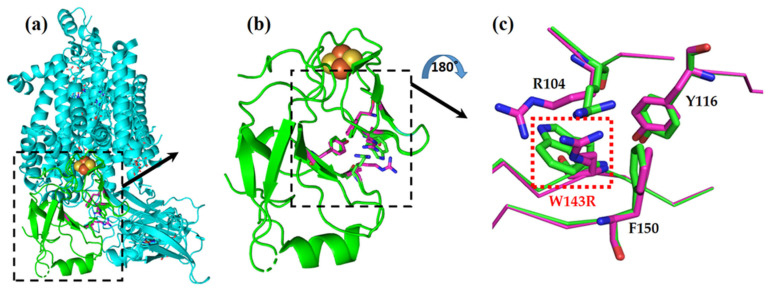
W143R mutation in Rieske protein of the cytochrome b6f complex, represented by two overlaid models: native (cyan and green) and the TSP9 mutant (magenta). (**a**) The cytochrome b6f complex from the membrane plane view, with the Rieske protein colored in green, and the rest of the subunits shown in cyan. (**b**) Close-up of the Rieske protein alone, from the membrane plane view. The 2Fe-2S cluster in orange and yellow can be seen in the proximity of the affected amino acids. (**c**) Close-up view of the W143R mutation site (red box) and the affected neighboring residues, arginine (R104), tyrosine (Y116), and phenylalanine (F150). The model view was rotated by 180 degrees to better show the mutation site.

**Figure 5 biomolecules-13-00537-f005:**
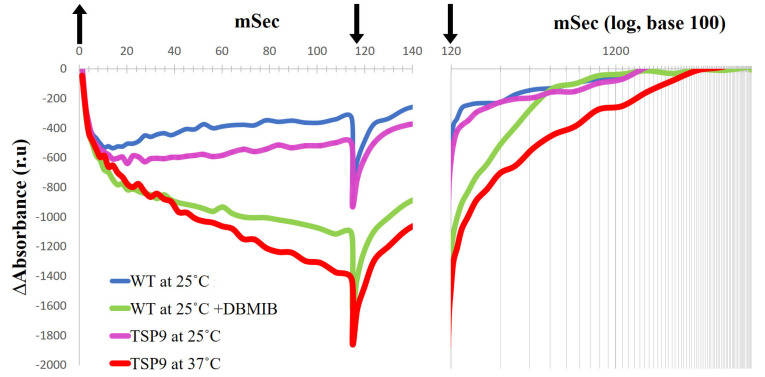
Absorbance records in relative units (ru) for light-induced P700 photooxidation (left panel) and P700+ dark-reduction (right panel, with time presented with a logarithmic scale) of the wild type (WT) and TSP9 *Chlamydomonas reinhardtii* cells (arrows: up, light ON; down, light OFF). P700 kinetics of the WT cells that were grown at the permissive temperature of 25 °C (blue line), and in the presence of 0.1 mM DBMIB (green line) are compared with P700 kinetics of TSP9 cells grown at the permissive (magenta line) and non-permissive (red line) temperatures. The chlorophyll concentration for each measurement was balanced to 0.1 mg/mL.

**Figure 6 biomolecules-13-00537-f006:**
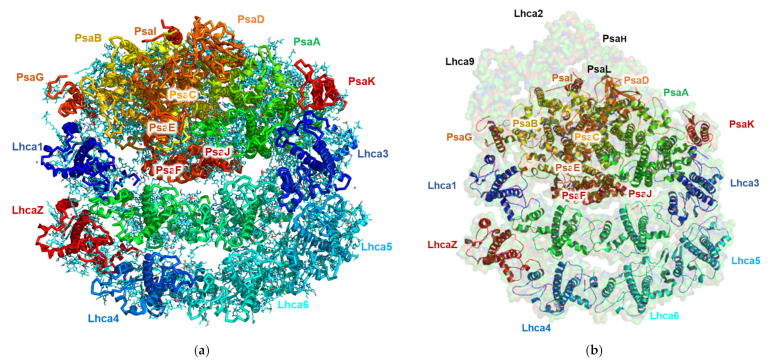
Structure of *Chlamydomonas reinhardtii* TSP9 PSI-8LHC (photosystem I (PSI)) at 2.42 Å (PDB 7R3K). The subunits are annotated according to their corresponding colors. (**a**) Cryogenic electron microscopy of TSP9-PSI-8LHC obtained from the 2.424 Å density map. The main chains are colored according to the subunits; the ligands are shown in cyan. (**b**) The TSP9 PSI-8LHC structure overlaid on surface representation at 80% transparency of the known structure for wild type PSI from *C. reinhardtii* (PDB 6JO5).

**Figure 7 biomolecules-13-00537-f007:**
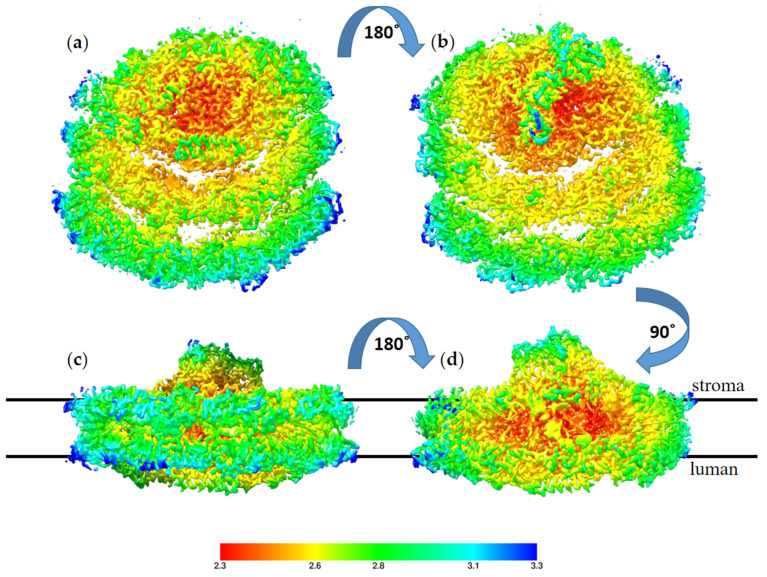
Local resolution of the TS9 PSI-8LHC (photosystem I (PSI)) structure, color coded as shown on the scale,
with red for the best resolution at 2.3 Å and blue for the resolution at 3.3 Å. The structure is shown from four angels: (**a**) view from the luminal side, (**b**) view from the stromal side, (**c**) view from the PsaD pole of the membrane plane, and (**d**) view from the LHC pole of the membrane plane. The stroma and lumen positions are marked for the membrane plane structures.

## Data Availability

The atomic coordinates have been deposited in the Protein Data Bank, with accession code 7R3K (TSP9-PSI-8LHC). The cryo-EM maps have been deposited in the Electron Microscopy Data Bank, with accession code EMD-14248 (TSP9-PSI-8LHC).

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
