# Peer review of "Structure of Photosystem I Supercomplex Isolated from a Chlamydomonas reinhardtii Cytochrome b6f Temperature-Sensitive Mutant"

_biomolecules, 2023, doi:10.3390/biom13030537_

Round 1
Reviewer 1 Report
The manuscript by Schwartz and colleagues describes the cryo-EM structure of the PSI complex isolated from the cells of C. reinhardtii with a W143R mutation in the petC gene. In these mutants, as the authors claim, the cytochrome b6f complex was absent when the cells were grown at 37°C, but not when grown at 25°C. The cryo-EM of PSI complexes isolated from the cells grown at 37°C showed that these complexes were depleted from several PSI subunits.
The manuscript can be published after a revision.
General comment.
The manuscript is extremely sloppily written. It is kind of a draft, not a final manuscript. It gives the impression that the person who wrote the manuscript hardly knows what a scientific article should look like. Therefore, it is recommended that experienced scholars check the manuscript before resubmission. Some corrections are suggested below, but there is no way to mention all the minor errors.
Major comments.
1) Introduction As a general rule, the Introduction section should include a description of relevant previous research. Here it should contain information on previous structural studies of PSI, particularly in C. reinhardtii, as well as on supercomplexes involving PSI and the cytochrome b6f complex. It is necessary to cover all relevant results obtained in different laboratories. In addition, the phenomenon of state transitions should be defined and described (unless the authors refuse from its consideration in the revised manuscript). So far, all this necessary information is missing from the Introduction; some of it can, however, be found in the Results section.
The first paragraph on global warming and the last paragraph on water structure preservation in various PSI preparations are irrelevant to the topic of the manuscript and can be deleted. Instead, the last paragraph of the Introduction section should start from a sentence “Here we show…” and briefly describe the main findings of the authors.
2) Methods and Results: Random mutagenesis by UV radiation can cause mutations in both nuclear DNA and chloroplast DNA. The latter possibility is not addressed in the manuscript. How do the authors know that no mutations have occurred in genes encoded by chloroplast DNA? Such mutations could also explain the observed effects. This point should be covered in the revised manuscript.
3) Figure 3 and related text: No information is provided on the expression of cytochrome b (petB) and subunit IV (petD) of the cytochrome b6f complex. This information is needed to understand whether the entire cytochrome b6f complex or only the Riske protein was absent from the membranes. Please provide this information.
4) Figure 5 and related text: someone who understands photosynthetic electron transport and its measurement should rewrite this section. As it stands, this part of the manuscript is unacceptable. The kinetic curves are hand-drawn with a marker pen; the output of the JTS-10 (Joliot-type spectrophotometer) is actually a line of dots. Even with these hand-drawn curves, it is unclear why turning off the light causes additional oxidation of P700; something seems to be wrong with the experimental setup.
Section 3.5.1:
5) Lines 402-409: This paragraph should be moved to the Introduction section; it contains no results.
6) Figure 6 and related text: the PsaH subunit is not indicated in the figure. Therefore, it is not clear where it is and how it could get lost. Generally, it is not easy to understand what is going on because there is no information in the "Introduction" section neither about the peripheral subunits of PSI, nor about the interaction between the PSI complex and cytochrome b6f. In fact, it has been shown that these two complexes form a supercomplex, particularly in C. reinhardtii; there is an article about this. In this case, there must be an interface between both complexes, where cytochrome b6f complex subunits can assist in the assembly of the peripheral PSI subunits, so that these peripheral subunits can be lost in the absence of the cytochrome b6f complex.
7) Lines 421-430: These sentences should be included in the "Introduction" section.
8) In the "Discussion" section, the data obtained should be considered in relation to the available functional and structural information about the functional and structural interactions between PSI and the cytochrome b6f complex. Instead, the "Discussion" section relates the results to state transitions. The problem is that there is no information in the manuscript about state transitions in the TSP9 mutant. Most likely, in the absence of the cytochrome b6f complex, mutant cells are fixed in one of two alternative states, but it is not clear which one. Therefore, the consideration of state transitions in connection with the presented data does not make sense and is perplexing.
Reviewer 2 Report
The manuscript by Schwartz et al. describes the characterisation of the TSP9 mutant. This mutant was isolated from a collection of 26,000 Chlamydomonas reinhardtii mutants screened for a temperature-sensitive photosynthesis phenotype. TSP9 was one of the seventeen TSP mutants (Temperature Sensitive Photosystem mutant) isolated using a negative selection strategy.
Physiological characterisation of the TSP9 mutant leads the authors to conclude that cytochrome b6f activity is somehow compromised in this mutant. Gene sequencing of cytochrome b6f subunits reveals a single T to C substitution in exon 3 of the petC gene, which replaces a highly conserved W residue for an arginine residue. Western blot analysis of the TSP9 mutant confirms low abundance of the petC protein at permissive temperature and full disappearance of the protein at restrictive temperature.
In the final part of the manuscript authors performs cryo-EM to elucidate the structure of the PSI complex in the TSP9 mutant grown at non-permissive temperature. The authors observe that the PSI subunits PsaL and PsaH, as well as the light harvesting subunits Lhca2 and Lhca9 are missing from the complex. Authors postulate that the structural changes of the PSI complex in the TSP9 mutant could be a response to prevent over-excitation of the PSI complex or a consequence of the production of reactive oxygen species by the PSII as the linear electron flow is inhibited due to the petC mutation.
I have several comments about the work presented:
1. I think that the work towards the characterisation of the TSP9 mutant (first part of the manuscript) is sound and the author’s conclusions are justified by the data. However, I think that some descriptions of the data are inaccurate: in line 306, authors say that "TSP9 showed normal levels of all four photosynthetic complexes at the permissive temperature (25C)…”. However, this is not the case because petC levels are very low in the TSP9 grown at 25C compared to the wild type strain (Figure 3)
2. Based on the sequencing and western blot data, authors conclude that a mutation in the petC gene is responsible for the photosynthesis phenotype of TSP9. In order to be certain about this conclusion, they should have attempted to transform the TSP9 mutant with a wild type copy of the petC gene and test if this effectively complements the mutation, abolishing the photosynthetic phenotype. Given that the protein disappears at non-permissive temperature there is no reason to think that the mutant petC gene is dominant.
3. I sympathise with the authors' effort to contextualise their work, but I think that the first paragraph of the introduction is unnecessary and out of the scope of this work.
4. Sections 3.1 and 3.2 of Results lack details about experiments and results. There is no justification of the experiments mentioned in the last paragraph of section 3.2 (lines 319 to 321), nor description of the results obtained. In addition, these experiments seem to show mixotrophic growth, while testing growth in phototrophic and heterotrophic growth would have been more appropriate for the characterisation of a photosynthetic mutant.
5. Unless I missed it, I cannot find any comment in the manuscript about the reduction in the levels of a second cytochrome b6f subunit, cytF, at non-permissive temperature, as well as the lower levels ATPb, subunit of the ATP synthase complex.
6. Authors refer to TAP+ and TAP- media in the manuscript. These media do not seem to correspond to the standard TAP, nor their composition is described in the manuscript.
7. A description of the genetic background of the strain mutagenised as well as the wild type control are missing from the material and method section.
8. About the cryo-EM data presented in this manuscript:
-The authors do not show or refer to the PSI structure of the wild type strain grown at non-permissive temperature. Therefore, they cannot strictly rule out that the change in the PSI structure is a response to the non-permissive temperature (37C).
-The authors observe changes in subunit composition of the PSI complex and they interpret these changes as a response to protect PSI from over excitation or PSII derived oxygen reactive species. Suga et al 2019 showed that the binding of the heterodimer Lhca2/Lhca9 to the PSI core is reversible. Disassociation of the heterodimer from PSI allows the formation of a cyt b6f-PSI supercomplex, which would work in support of circular electron flow for ATP generation (Steinbeck et al. 2018, PNAS 115:10517). Given the unstable binding of the Lhca heterodimer, how can the authors be sure that these subunits were not lost during the purification of the complex?
-Does Suppl. Figure 8 show whether Lhc2/Lhca9 and PsaL are actually missing? It would be helpful if the bands were labeled.
-I cannot find justification for the use of a petC temperature sensitive mutant to study PSI complex dynamics, nor that it has added anything new to its understanding. C. reinhardtii is an exceptional model organism for the study of photosynthesis, and there is already a large collection of mutants affected in every single photosynthetic complex. In addition, there are several compounds that inhibit photosynthesis at different points of the process, facilitating the study of dynamic responses. The generation of temperature sensitive mutants in this process seems interesting, but in this particular example I do not think they have been exploited to add novel mechanistic knowledge about PSI.
-
Round 2
Reviewer 1 Report
The authors have essentially improved the manuscript and have addressed most of the comments.
Still, some further revision is needed.
Major comment to Section 3.4. Effect of TSP9 mutation on ET (electron transport).
Concerning comment 4 [Figure 5 and related text: someone who understands photosynthetic electron transport and its measurement should rewrite this section. As it stands, this part of the manuscript is unacceptable. The kinetic curves are hand-drawn with a marker pen; the output of the JTS-10 (Joliot-type spectrophotometer) is actually a line of dots. Even with these hand-drawn curves, it is unclear why turning off the light causes additional oxidation of P700; something seems to be wrong with the experimental setup] the authors wrote:
“Response 4: while the JTS-10 visual output is indeed a line of dots, these dots are also given in quantitative levels as a function of time. The curves shown in figure 5 were drawn by excel using the quantitative output of the JTS-10. The original curves visualized by the JTS are attached to confirm the data presented. Moreover, another repetition of this experiment is also attached in order to show these curves are not a 1-time occurrence. Since the JTS measurements were performed in vivo on full cells containing all involved proteins, the additional oxidation of p700 seen when the light was turned off is assumed to represent the electron returning from p700 to oxidized PC. When the measurements are performed in vitro on isolated proteins with the specific electron carriers PMS and ascorbic acid, these jumps in the curves disappear. the text has been revised according to the notes given here in order to clarify the curves seen.”
At this point some further work is needed. Fig.5 should be replaced by original data (dots + lines), which were provided by the authors in the response. These data contain measurements under five different conditions (and not under four, as in the original manuscript); they, therefore, contain more information. The Y axis should be labeled in true units of optical density, currently there are some arbitrary units.
The text of this section should still be reviewed by an expert in electron transfer. The text is unclear and wrong in many places. Specifically, FNR is not located downstream of P700 in the ET chain because an electron cannot reach FNR from P700 without being photoexcited. Generally, the terms “upstream” and “downstream” can be misleading when applied to photosynthetic electron transfer chains, better to avoid them. Also, it is unlikely that an electron can return from P700 to PC, as authors suggest in the revised manuscript, because the midpoint redox potential of P700 is much higher than that of PC. Furthermore, the amplitude of fast absorbance changes in response to switching the light off do not depend on whether P700 was oxidized by light (DBMIB, TSP9 at 37 C) or not. Most likely these fast absorbance changes are unrelated to the electron transfer around P700 (perhaps, electrochromism of chlorophyll of some kind or a bug in the experimental setup?).
Minor comment: The English language should be checked, particularly in relation to description of the subunits of PSI that lack in the structure. It seems not quite appropriate to write about them as “missing” (who is missing them?) or “omitted” (by whom?).
Reviewer 2 Report
I appreciate the authors's effort to answer some of the points that I raised in the first round of revision, even though my opinion was that the manuscript did not have the merits for publication in Biomolecules. Some of the authors' answers have satisfactorily addressed my points (points 1, 2, 3, and 5). In their letter of reply, the authors say that they have addressed point 6 in the supplementary data; however I could not find the supplementary Materials section that they referred, nor the extra labelling in Figure S8. On the other hand, their reply to point 4 is still not addressing the original criticism, and in any case they think it is not worth to address it in the actual manuscript.
However, the most sticky point in my opinion is the interpretation of the cryo-EM data (point 8), which constitutes a big part of the article. With the data that they present, the authors cannot confidently conclude that the changes in the PSI structure that they describe is the result of the TSP9 mutation or an artefact of the incubation at 37C. The absence of a number of subunits can be even the result of incomplete purification of the complex. In addition, I am still wondering about the usefulness of these photosynthetic temperature sensitive mutants to answer questions regarding photosynthesis dynamics. Their reply to point 8.4 does not seem to help.
